# Navigating Uncertainty: Enhancing Markowitz Asset Allocation Strategies through Out-of-Sample Analysis

Vijaya Krishna Kanaparthi

Microsoft Corporation, 7000 State Highway 161,Building LC1, Irving, TX 75039, USA; datalivesite@gmail.com

**Abstract:** This research paper explores the complicated connection between uncertainty and the Markowitz asset allocation framework, specifically investigating how mistakes in estimating parameters significantly impact the performance of strategies during out-of-sample evaluations. Drawing on relevant literature, we highlight the importance of our findings. In contrast to common assumptions, our study systematically compares these approaches with alternative allocation strategies, providing insights into their performance in both anticipated and real-world out-of-sample events. The research demonstrates that incorporating methods to address uncertainty enhances the Markowitz framework, challenging the idea that longer sample periods always lead to better outcomes. Notably, imposing a short-sale constraint proves to be a valuable strategy for improving the effectiveness of the initial portfolio. While revealing the complexities of uncertainty, our study also highlights the surprising resilience of basic asset allocation approaches, such as equally weighted allocation, which exhibit commendable performance. Methodologically, we employ a rigorous out-of-sample evaluation, emphasizing the practical implications of parameter uncertainty on asset allocation outcomes. Investors, portfolio managers, and financial practitioners can use these insights to refine their strategies, considering the dynamic nature of markets and the limitations internal to the traditional models. In conclusion, this paper goes beyond the theoretical scope to provide substantial value in enhancing real-world investment decisions.

**Keywords:** expected events; equally weighted allocation; Markowitz asset allocation; out-of-sample events

**JEL Classification:** C38; C45; D81

## 1. Introduction

According to an ancient Talmudic proverb, people should set aside three equal portions of their money: one for land, one for business, and one for storage. The concept of allocating funds among various assets is referred to as "asset allocation". There is a wide range of professional viewpoints on the best approach to accomplish this with no consensus, despite the fact that many have tried to reach one. A scientist by the name of Markowitz made a substantial addition to this field in 1952 [1]. In 1952, the Markowitz [2] model was introduced and had a significant effect on portfolio management and altered overall investment strategies. Developed by Markowitz, the model put forth the revolutionary idea of the risk–return trade-off. It focused on allocating assets by considering covariance, standard deviation, and projected returns, aiming to optimize portfolios for the least amount of risk at a pre-determined expected return. This groundbreaking approach set the foundation for modern portfolio theory, shaping investment practices for many years. To determine the optimal approach to allocating the money among assets, he employed concepts such as covariances—measures of how the returns of various assets relate to one another—and standard deviation, which is a measure of risk. He discovered that the least risky portfolio for a given projected return is the optimal one. The estimates utilized for variances, covariances, and expected returns, however, have an impact on this approach.

Stated differently, the precision of these approximations is critical; otherwise, the model may not perform as intended in practice. "Parameter uncertainty"[3] is another term for this problem of ambiguous estimations. This raises a few crucial queries:

- How can the Markowitz framework's uncertainty be explained?
- Does the model perform better when uncertainty is taken into account?
- How do these optimized allocations stack up against alternative tactics?
- Can the model do better with more data?
- Is it feasible to make a trustworthy performance prediction?
- Does performance improve when there are limitations on allocation?

Since Markowitz's pioneering work in 1952, methods for dividing assets have evolved to address the challenge of uncertainty. Notable advancements involve improvements in risk management models that surpass covariances and standard deviation by considering additional factors. The application of advanced statistical methods and technological innovations enables more precise risk assessments. Researchers have explored Bayesian methods and utilized machine learning [4–17] to enhance parameter estimations. Additionally, an increasing emphasis on behavioral finance has yielded insights into investor psychology, influencing decisions on asset allocation. These post-Markowitz developments present a dynamic landscape, highlighting the ongoing search for strategies capable of effectively navigating uncertainty—a key focus of this study.

However, despite decades of exploration, effectively managing uncertainty in asset allocation remains a critical and challenging aspect. This study aims to fill the existing research gap by investigating various approaches that deal with parameter uncertainty within the Markowitz framework. The key questions posed—concerning developing an explanation of uncertainty, enhancing model performance, comparing alternative tactics, considering the impact of data abundance, evaluating the reliability of performance predictions, and examining the influence of allocation constraints—highlight the gaps in current understanding. By comparing these approaches with established models and exploring less explored factors, this research contributes to advancing comprehension and practical application in the field. Therefore, in this study, I investigate different approaches to asset allocation that try to deal with this ambiguity about the parameter. I compare these options against the original model as well as alternative approaches such as the "1 over N" strategy. I consider both the real-world performance and the expected performance. I also discuss a topic that has not received much attention: short-sale limits on assets. I also suggest a model extension that takes parameter uncertainty into account. To be sure that the results are trustworthy, I run our analyses on two distinct sets of data. Recognizing the already available research on optimizing portfolios with short-sale constraints, my goal is to add detailed insights within the broader context of this study.

The paper is as follows: In Section 2, the Markowitz model's theoretical foundation and constraints are covered, whereas Section 3 investigates model extensions for stronger allocations based on related works. The empirical analysis is covered in the Section 4. I give the findings and methodologies, as well as the sources of the data, in Sections 5 and 6. I evaluate the various asset allocation approaches in both simulated and actual environments in Section 7. Section 8 presents a discussion section to discuss the limitations and the challenges faced. The study is concluded in Section 9, which discusses the analysis and their results.

## 2. Background: The Mean-Variance Model, Criticism, and Its Extensions

### 2.1. Asset Allocation According to Markowitz

The classic asset allocation problem in finance involves an investor seeking to distribute their initial wealth, denoted as $W_0$, among N risky assets, with returns assumed to follow a normal distribution characterized by expected rates of return ($\mu$) and a variance–covariance matrix ($\Sigma$). The investor's goal is to determine the optimal portfolio ($\pi$) that maximizes their expected final wealth, given their preferences quantified by a utility function, typically the Constant Absolute Risk Aversion (CARA) utility function. The CARA utility function,

$U(W) = -e^{-kw}$, reflects the investor's profit-seeking and risk-averse nature. The CARA parameter, *k*, represents the investor's risk tolerance. A positive *k* implies an upward-sloping utility function, becoming less steep as wealth increases, aligning with the investor's preference for higher wealth and aversion to risky prospects. Assuming a normal distribution of returns, the final wealth (*W*) is also normally distributed with mean $\mu(\pi)$ and variance $\sigma^2(\pi)$. The optimization problem for the rational investor involves maximizing expected satisfaction or utility, given by $E[U(W)]$, leading to the mean-variance criterion:

$$\max_{\pi} \pi^T \mu - \frac{\gamma}{2} \pi^T \Sigma \pi \tag{1}$$

Here, $\gamma$ represents the investor's risk aversion, with higher values indicating stronger aversion to variance. The optimization, known as the mean-variance criterion, is crucial for portfolio selection, considering both expected returns and risk. In a one-fund setting, where investors can only invest in risky assets, the optimization simplifies. However, in a two-fund setting, a risk-free asset is introduced, expanding the model. The investor now allocates wealth between the risky portfolio and the risk-free asset, considering the risk–return trade-off. In the mean-variance framework, the investor seeks to balance risk and return to achieve the most favorable outcome. The utility function's exponential form reflects diminishing marginal utility of wealth, capturing the investor's aversion to risk. The model helps investors make informed decisions on portfolio allocation, balancing risk and reward based on their risk preferences. The analysis also highlights the importance of the risk aversion parameter, $\gamma$, in the decision-making process. A higher $\gamma$ implies a lower tolerance for risk, leading to a more conservative investment strategy. The mean-variance criterion provides a quantitative approach to portfolio optimization, aligning with the principles proposed by Markowitz. Essentially, there is a solid mathematical argument proposing that solving $\max\{E[U(W)]\}$ is equivalent to solving the optimization task illustrated in Equation (1). This connection underlines that optimizing the CARA utility function efficiently aligns with the conventional mean-variance criterion put forth by Markowitz. This again reinforces the theoretical foundation of the portfolio optimization process.

### 2.2. Mean-Variance Efficient Portfolios in a One-Fund Setting

The mean-variance framework, introduced by Harry Markowitz in 1952, offers a solution for optimal asset allocation in the absence of a risk-free asset. At its core is the concept of the efficient frontier, a curved line on a risk–return graph that represents the optimal balance between expected portfolio return and standard deviation. This model capitalizes on the benefits of diversification by combining assets with less than perfect correlation, thereby reducing idiosyncratic risks. The goal is to determine the optimal weights for each asset in the portfolio to minimize overall risk. For a given expected return, denoted as $\mu$, the mean-variance efficient portfolio aims to minimize the portfolio's risk, subject to constraints. The Lagrange function is employed to solve the optimization problem, incorporating Lagrange multipliers. The optimal portfolio weights are determined using specific constants. These weights represent the least risky allocation among all possible portfolios with the same expected return. In essence, the mean-variance efficient portfolio provides a systematic approach to construct portfolios that offer the best risk–return trade-off. It remains a foundational concept in modern portfolio theory, guiding investors in constructing well-balanced and diversified investment portfolios.

### 2.3. Optimal Portfolio under CARA Utility in a One-Fund Setting

The following approach is taken to the problem of investment optimization for an investor who follows Equation (1)'s explanation of the CARA utility function. First, we define the Lagrangian function.

$$L = \pi^T \mu - \frac{\gamma}{2} \pi^T \Sigma \pi + \alpha \left(1 - \pi^T 1\right) \tag{2}$$

We then set the first-order conditions:

$$\frac{\partial L}{\partial \pi} = \mu - \gamma \Sigma \pi - \alpha 1 = 0 \tag{3}$$

and

$$\frac{\partial L}{\partial \alpha} = 1 - \pi^T 1 = 0 \tag{4}$$

Next, we simplify the $\pi$ yields:

$$\pi = \frac{1}{\gamma} \Sigma^{-1} \left( \mu - \frac{B - \gamma}{C} 1 \right) \tag{5}$$

As illustrated in Equation (5), B ($B$ represents the risk aversion coefficient and is calculated as $B = \mu^T \Sigma^{-1} 1$, where $\mu$ is the mean vector and $\Sigma$ is the covariance matrix of returns) and C ($C$ represents the curvature of the investor's utility function and is calculated as $C = 1^T \Sigma^{-1} 1$, where 1 is a vector of ones)are showcased as the basic parameters representing distinct quantities within an optimization framework. The efficient frontier and CARA-optimized asset allocations can now be included in the risk–return as shown in Table 1. We looked at monthly results to calculate the risk and return. The analysis in Section 2 then makes use of these monthly return figures. We utilize $\mu$ and $\Sigma$ to symbolize them. As can be seen in Table 1, both investors' upper computed allocations show investments that are in line with their preferences. Since both utility functions cross the efficient frontier tangentially and further upward shifts are not possible, these allocations are regarded as optimal, as shown in Tables 2 and 3. Moreover, it was found that the individual properties of individual assets are located in the inefficient region beneath the efficient frontier. Purchasing just one asset reduces the efficiency of the investment because it does not take advantage of diversity, as shown in Figure 1.

**Table 1.** Annualized $\mu$-vector and $\Sigma$ matrix of eight assets chosen.

| | $\mu$ Annualized | Variance–Covariance Matrix of Annualized Returns $\Sigma$ | | | | | | | |
| --- | --- | --- | --- | --- | --- | --- | --- | --- | --- |
| | | Asset 1 | Asset 2 | Asset 3 | Asset 4 | Asset 5 | Asset 6 | Asset 7 | Asset 8 |
| **Asset 1** | 2.30% | 6.81% | 3.03% | 2.36% | 2.32% | 1.79% | 2.71% | 3.60% | 2.02% |
| **Asset 2** | 6.50% | 3.03% | 4.96% | 2.96% | 2.41% | 2.04% | 2.70% | 3.72% | 1.99% |
| **Asset 3** | 8.22% | 2.36% | 2.96% | 3.35% | 2.01% | 1.75% | 2.44% | 2.88% | 1.85% |
| **Asset 4** | 5.79% | 2.32% | 2.41% | 2.01% | 3.94% | 2.30% | 2.52% | 2.56% | 1.59% |
| **Asset 5** | 6.64% | 1.79% | 2.04% | 1.75% | 2.30% | 2.40% | 2.10% | 2.10% | 1.21% |
| **Asset 6** | 5.14% | 2.71% | 2.70% | 2.44% | 2.52% | 2.10% | 4.40% | 3.07% | 1.92% |
| **Asset 7** | 6.15% | 3.60% | 3.72% | 2.88% | 2.56% | 2.10% | 3.07% | 5.14% | 2.13% |
| **Asset 8** | 7.32% | 2.02% | 1.99% | 1.85% | 1.59% | 1.21% | 1.92% | 2.13% | 4.50% |

**Table 2.** Asset allocation in a one-fund setting.

| Asset $i$ | 1 | 2 | 3 | 4 | 5 | 6 | 7 | 8 |
| --- | --- | --- | --- | --- | --- | --- | --- | --- |
| $\pi_i, \gamma = 1$ | −92.35% | −27.61% | 173.13% | −32.26% | 96.97% | 75.67% | 12.57% | 45.22% |
| $\pi_i, \gamma = 2$ | −43.50% | −15.75% | 97.92% | −17.70% | 82.01% | −38.93% | 1.91% | 33.83% |

**Table 3.** Portfolio characteristics in a one-fund setting.

|  | $\gamma = 1$ | $\gamma = 2$ |
|---|---|---|
| $E[r]$ | 15.07% | 11.05% |
| $\sigma$ | 31.71% | 20.04% |

**Figure 1.** Efficient frontier and mean-variance-optimized allocations for $\gamma = 1, 2$.

*2.4. The Global Minimum-Variance Portfolio*

Portfolios recognized for their distinctive qualities are displayed on the efficient frontier. The minimum-variance portfolio is one example of this type of portfolio, distinguished by its allocation, which reduces variation in comparison to all other portfolios. Equation (6) provides an illustration of this optimization puzzle.

$$\min_{\pi} \pi^T \Sigma \pi \tag{6}$$

$$s.t. \ \pi^T 1 = 1$$

The expression for $\pi_{min}$ is obtained by first decreasing the variability, and then substituting the expected portfolio return.

$$\pi_{min} = \frac{1}{C}\Sigma^{-1}1 = \frac{1}{1^T\Sigma^{-1}1}\Sigma^{-1}1 \tag{7}$$

One notable characteristic of the minimal variance portfolio is its detachment from the expected return vector. Its computation is independent of expected returns because it depends only on the variance–covariance matrix. In this portfolio, assets with lower standard deviations and/or little association with other assets are more valuable than those with contrary characteristics. To determine the minimum-variance allocation's expected return and variance, Equations (8) and (9) must be used.

$$\mu_{min} = \frac{B}{C} \tag{8}$$

$$\sigma^2_{min} = \frac{1}{C} \tag{9}$$

The weights obtained for the minimum-variance allocation are listed in Table 4 using the parameters from Table 1. Investment in the minimal variance portfolio is not the best option for risk aversions of $\gamma = 1$ and $\gamma = 2$, as indicated by the risk–return plot, which shows an expected return of 7.02% and a portfolio standard deviation of 14.15%. Allocating within the gray zones for each individual risk aversion level will yield more utility. In a CARA utility situation, this holds true for all risk aversions since, as Figure 2 illustrates, the efficiency frontier's magnitude at the minimal variance allocation is indefinite.

**Table 4.** Minimum-variance portfolio weights.

| Asset $i$ | 1 | 2 | 3 | 4 | 5 | 6 | 7 | 8 |
|---|---|---|---|---|---|---|---|---|
| $\pi_i$ | 5.35% | −3.90% | 22.72% | −2.74% | 67.05% | −2.19% | −8.74% | 22.74% |

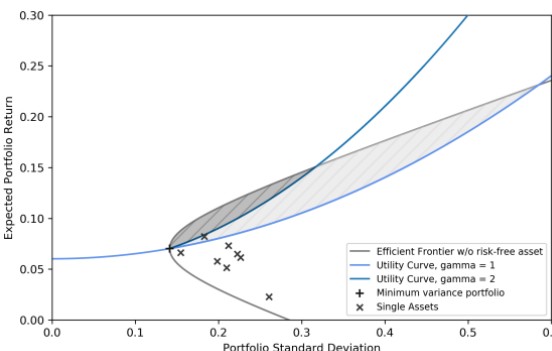

**Figure 2.** Risk–return diagram depicting the minimum-variance portfolio.

## 2.5. Mean-Variance Efficient Portfolios in a Two-Fund Setting

The extension of the original Markowitz approach to include a risk-free asset allows investors to optimize their portfolio allocation by considering a combination of risky assets and risk-free security. The wealth at the end of the investment period ($W$) is determined by the initial wealth ($W_0$), the risk-free rate ($r_f$), and the allocation to the risky portfolio ($\pi > \mu$). The optimal asset allocation involves determining the fraction invested in the risk-free asset and the remaining fraction in the risky portfolio. For an investor with CARA preferences, the mean-variance criterion is applied to maximize expected utility. The optimal weight ($\omega_p$) in the risky portfolio is established by considering the return ($r_p$) and standard deviation ($\sigma_p$) of the risky portfolio, along with the risk-free rate ($r_f$). The investor aims to maximize the expected utility by solving the optimization problem shown in Equation (10).

$$\max_{\omega_p} \omega_p r_p + (1 - \omega_p) r_f - \frac{\gamma}{2} \omega_p^2 \sigma_p^2 \tag{10}$$

The First-Order Condition (FOC) for this optimization yields the optimal weight in the risky portfolio, represented by Equation (11).

$$\omega_p^* = \frac{r_p - r_f}{\gamma \sigma_p^2} \tag{11}$$

Equation (11) demonstrates that the amount invested in the risky asset is inversely proportional to the investor's risk aversion coefficient ($\gamma$) and directly related to the excess returns of the risky assets over the risk-free return. This result is intuitive; risk-averse investors allocate less to risky assets and more to the risk-free asset, emphasizing the trade-off between risk and return in portfolio optimization. Figure 3 shows that the combination of risky and risk-free assets results in multiple possible portfolios, represented by dashed lines. Each dashed line signifies a different allocation of wealth between the risky and risk-free assets. The tangent line with the highest utility, depicted by the blue curve, represents the optimal portfolio for an investor with CARA preferences. In this hypothetical scenario, the investor would invest 101% in the risky asset and borrow 1% (negative allocation) from the risk-free asset, showcasing the potential for leveraging strategies.

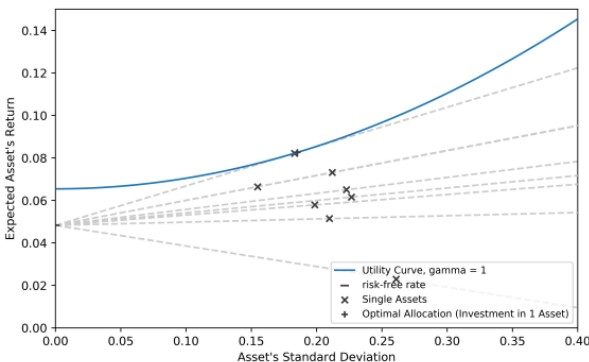

**Figure 3.** Risk–return diagram of two-fund combinations.

The Global Maximum Sharpe Ratio Portfolio/Tangency Portfolio

Transitioning from the restricted case, as depicted in Figure 3, to a mean-variance optimization setting introduces a pivotal concept: the tangency portfolio. This portfolio, crucial in portfolio theory, represents the optimal allocation into risky assets when combined with the risk-free asset, maximizing utility for all investors. The fraction invested in the tangency portfolio and the risk-free asset is determined by Equation (11), emphasizing the importance of risk aversion and excess returns in the decision-making process. A distinctive feature of the tangency portfolio is its possession of the highest Sharpe ratio among all portfolios. The Sharpe ratio [18], defined as the excess return of an asset per unit of risk, is a widelyused metric for assessing risk-adjusted performance. In the context of the tangency portfolio, the Sharpe ratio is represented by the slope of the straight-line tangent to the efficient frontier of risky assets. Mathematically, it is given by Equation (12).

$$SR = \frac{r_i - r_f}{\sigma} \qquad (12)$$

where $r_i$ is the return of asset $i$, $r_f$ is the risk-free rate, and $\sigma$ is the standard deviation of return. The weights of the tangency portfolio in risky assets can be determined using Equation (13).

$$\pi_{tan} = \frac{1}{B - C_{r_f}} \Sigma^{-1} \left( \mu - r_f 1 \right) \qquad (13)$$

The first term in Equation (13) is a scalar, and the second term, $\Sigma^{-1}(\mu - r_f 1)$, is an $N \times 1$ vector, accounting for the correlation structure among assets. This expression, similar to the Sharpe ratio, guides the allocation of assets in the tangency portfolio. Considering an investor with CARA preferences, the weight vector for the tangency portfolio, including risk aversion, is expressed according to Equation (14).

$$\pi^*{}_{tan} = \frac{\mu_{tan} - r_f}{\gamma \sigma^2_p} \cdot \frac{\Sigma^{-1} \left( \mu - r_f 1 \right)}{B - C_{r_f}} \qquad (14)$$

Equation (14) shows that the weight in the tangency portfolio is influenced by risk aversion ($\gamma$) and the correlation structure among assets. The complementary weight in the risk-free asset is $1 - \Sigma \pi^*{}_{tan}$. Assuming a risk-free rate of 4.81%, weights for the tangency portfolio composition were obtained as shown in Table 5. For two example investors with risk aversions of one and two, the optimal investment involved investing a weight of $\omega_p$ in the tangency portfolio and the remainder in the risk-free asset is shown in Table 6. The resulting efficient frontier, incorporating the risk-free asset, illustrated the highest level of utility any investor could achieve. The weights of 109.66% and 54.83% in the risky asset for risk aversions of one and two, respectively, demonstrate how investors with different risk preferences allocate their wealth differently between the tangency portfolio and the risk-free asset. Comparing this figure to the previous restricted case (Figure 3), the tangency line

now represents the highest level of utility achievable according to Figure 4. Furthermore, in comparison to the efficient frontier in the one-fund setting (Figure 1), the inclusion of the risk-free asset elevates the overall utility level, emphasizing the benefits of diversification and risk-free assets in portfolio optimization.

**Table 5.** Tangency portfolio weights $\pi_{tan}$.

| Asset $i$ | 1 | 2 | 3 | 4 | 5 | 6 | 7 | 8 |
|---|---|---|---|---|---|---|---|---|
| $\pi_i$ | −83.74% | −25.52% | 159.87% | −29.66% | 94.34% | −69.19% | 10.69% | 43.22% |

**Table 6.** Optimal two-fund allocation for $\gamma$ = 1, 2.

| | $\gamma = 1$ | $\gamma = 2$ |
|---|---|---|
| $\omega_p$ | 109.66% | 54.83% |
| $E[r]$ | 15.29% | 10.06% |
| $\sigma_p$ | 32.34% | 16.17% |

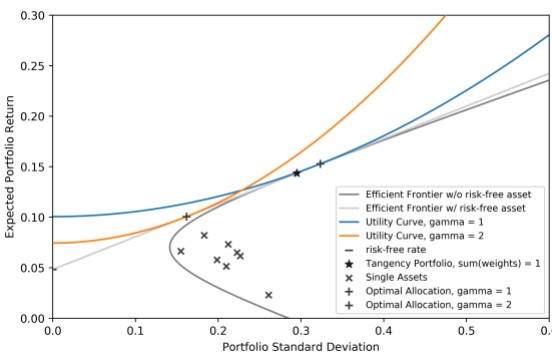

**Figure 4.** Risk–return diagram including capital allocation line.

## 3. Related Works

The mean-variance optimization framework, pioneered by Harry Markowitz in 1952, has long been a cornerstone in modern portfolio theory. This approach, while theoretically appealing, faces challenges in practical applications due to the sensitivity of the model to input parameters, primarily the expected return vector ($\mu$) and the variance–covariance matrix of asset returns ($\Sigma$). This sensitivity poses significant issues for real-world asset allocation decisions. Markowitz's mean-variance optimization offers several advantages and features, as highlighted by [19,20]. It provides a structured framework for considering essential portfolio constraints, leverages diversification based on correlation structures, optimally uses available information, and allows for quick calculation of asset allocations, facilitating adaptability to changing market conditions. However, the classical mean-variance model assumes knowledge of true input parameters, which is practically unattainable. This limitation, coupled with the model's sensitivity to parameter estimates, has led to its lack of acceptance among investment professionals. The out-of-sample performance of the mean-variance model has been notably poor, prompting extensive research into understanding the reasons behind its underperformance and proposing improvements. Ref. [19] coined the term "estimation-error maximizer" to describe mean-variance portfolios' sensitivity to parameter uncertainty. Ref. [21] suggest that this could explain why the model is not widely applied in practice. The parameter uncertainty arises from the reliance on estimates for ($\mu$) and ($\Sigma$), and researchers have explored the impact of such uncertainty on portfolio composition and financial outcomes. Researchers like [22] have focused on the sensitivity of mean-variance portfolios to changes in the expected return vector. They found that for unconstrained portfolios, even small changes in expected returns could lead to significant shifts in portfolio weights, reducing the benefits of diversification. This sensitivity becomes

more pronounced as the number of assets increases, indicating potential instability in solutions. Refs. [19,23] highlighted issues with high values in the variance–covariance matrix components, leading to instability in the inverse matrix ($\Sigma^{-1}$). A high condition number of $\Sigma$ indicates instability, particularly problematic when the availability of historical data is limited. Researchers have proposed shrinkage estimators to mitigate this instability and provide more-robust variance–covariance matrix estimates. A Monte Carlo simulation illustrates the impact of parameter uncertainty on optimal asset allocation. This simulation has revealed that estimated portfolio characteristics tend to overstate expected returns and understate risk, leading investors to believe that they are making optimal decisions than they actually are. The discrepancy between estimated and true parameters emphasizes the importance of addressing parameter uncertainty in portfolio optimization. Research has also shown that mean-variance portfolios can underperform simpler strategies, such as equally weighted portfolios, in out-of-sample scenarios. Refs. [1,13] found that estimation errors erode the advantages of diversification, and the out-of-sample Sharpe ratios of simpler strategies can surpass those of mean-variance portfolios. Refs. [3,24] investigated the utility loss caused by suboptimal portfolios resulting from parameter estimation errors. They observed larger losses concerning expected return estimates compared to variances or covariances. Ref. [3] provided mathematical proof for utility loss, with the impact being more significant for expected return estimates. Furthermore, concerns about the assumptions underlying the mean-variance model, such as normal distribution of returns and a negative exponential utility function, contribute to its low acceptance. Traditional portfolio managers may resist adopting a quantitative investment process, contributing to the model's limited adoption in practice. In conclusion, the mean-variance optimization model, while theoretically elegant, faces challenges in real-world applications due to its sensitivity to parameter estimates and assumptions. Researchers continue to explore ways to enhance its robustness, acknowledging the importance of addressing parameter uncertainty and considering alternative portfolio strategies that may perform better in practical scenarios.

## 4. Empirical Study: Improving the Mean-Variance Setting

### 4.1. Improving Parameter Estimation

This section discusses challenges associated with classical mean-variance optimization models and explores alternative approaches to improve parameter estimation for portfolio optimization. Despite the statistical soundness of Maximum Likelihood Estimators (MLE), their poor out-of-sample performance prompts a search for more robust methods. The traditional Bayesian approach, based on the works of [3,25], is introduced. It involves deriving expected utility through an integral over the product of utility given a return and the probability of its realization, with the probability depending on a set of parameters denoted as $\Theta$. The Bayesian approach assumes that investors have a prior belief about these parameters, combining this belief with historical returns to derive a conditional prior. The resulting expected utility is expressed as an integral involving both the utility given a return and the product of the conditional probability and the likelihood. The Bayesian approach combines historical data with prior beliefs to influence wealth allocation. Despite the estimation risk represented in the likelihood acquired through Bayesian estimation, investors remain neutral to parameter uncertainty. Ref. [3] provide an analytic proof for the expected utility loss induced by parameter estimates. They introduce the expected utility loss function, or risk function, defined as the difference between the true and estimated weight vectors. The utility function allows for the comparison of losses among investors with different risk aversions. The discussion extends to mean-variance efficient portfolios in a three-fund setting, considering scenarios with only risky assets or both risky assets and a risk-free security. The challenge is highlighted where the expected utility arising from an allocation may be lower than the utility obtained when weights are derived based on certain parameters. Ref. [3] propose a three-fund allocation as an alternative to mitigate uncertainty issues, incorporating a global minimum-variance portfolio. The allocation

involves optimizing constants *c* and *d* to maximize the expected certainty of the equivalent return, as shown in Equation (15).

$$\hat{\pi} = \hat{\pi}(c, d) = \frac{1}{\gamma}\left(c\Sigma^{-1}\left(\hat{\mu} - r_f 1^T\right) + \left(d\Sigma^{-1} 1\right)\right) \tag{15}$$

Another Bayesian approach, proposed by [26], focuses on improving the estimate of the variance–covariance matrix. The simple MLE is consistent but not necessarily robust. The shrinkage approach proposed by ref. [26] based on a structured estimator like the constant correlation model, aims to make covariance matrix estimates more robust. The approach involves a shrinkage coefficient to balance the structured estimator with the sample covariance matrix, pulling extreme values towards the central diagonal. The optimal shrinkage coefficient is determined by minimizing the expected distance between the shrinkage estimator and the true covariance matrix. The resulting shrunk matrix is a combination of the structured estimator and the sample covariance matrix.

### 4.2. Asset Allocation in a Downside Risk Setting

Research by ref. [27] challenges the suitability of conventional risk measures, such as variance and standard deviation, in the mean-variance setting when faced with multiple constraints. The traditional reliance on variance as a risk measure is rooted in the assumption of normal distribution of returns. However, ref. [27] highlights the limitations of these measures, emphasizing that variance may not adequately represent an investor's view on risk, especially in the presence of constraints. In the mean-variance setting, the assumption of normal distribution implies that the only relevant dispersion measure is the variance. Nonetheless, some investors may be more concerned with the dispersion of returns around a specific target return, deviating from the mean return assumed in traditional mean-variance settings. This observation prompts the exploration of asymmetric risk measures that focus on specific portions of the return distribution, particularly the left tail in a downside risk context. Ref. [27] introduces the concept of Lower Partial Moments (LPMs) as a more suitable measure of risk in this framework. LPMs are asymmetric risk measures that consider only a selected portion of the return distribution, specifically the left tail. These measures are designed to capture an investor's perspective on risk by giving greater weight to losses than gains, aligning with behavioral finance theories. The general form of *n*-LPM is expressed as the sum of the probability-weighted deviations below a specified target rate, $\tau$. The optimization problem associated with LPM involves constructing an efficient portfolio within a downside risk framework. The focus is on improving the risk/return tradeoff while minimizing the target semi variance. The optimization problem can be expressed mathematically as the minimization of the sum of the probability-weighted squared deviations of returns below the target, as shown in Equation (16).

$$min_\pi \sum_\tau P_r(r_p < \tau) \cdot (\tau - r_p) \text{ subject to constraint } \sum_i \pi_i > 1 \tag{16}$$

Here, $\pi$ represents the portfolio weights, $r_p$ denotes the returns of the portfolio, $\tau$ is the target rate, and $P_r(r_p < \tau)$ is the probability of returns being below the target. The optimization problem seeks to find the optimal portfolio weights that minimize the target semi variance. It is crucial to note that, unlike the mean-variance setting, there is no closed-form solution to this optimization problem. Therefore, numerical solvers are employed to determine the optimal portfolio weights, $\pi^*$.

### 4.3. Introducing Ambiguity Aversion into a Mean-Variance Setting

Ref. [25] propose an extension to the traditional mean-variance model to address the challenge of incorporating investor aversion to parameter uncertainty, specifically focusing on the estimation of expected returns. This extension introduces the concept of ambiguity aversion, where an investor expresses a preference for knowing the probability distribution of an outcome over remaining uncertain. The distinction between ambiguity aversion and

risk aversion is crucial, with the former representing aversion to unknown distribution and the latter to the actual risk associated with a decision. The theoretical foundation of ambiguity aversion is laid out, drawing on the work of [28], which demonstrates that individuals tend to prefer situations where they can calculate probabilities rather than guessing them, even when facing uncertainty about the probability distribution. Ambiguity aversion in the context of [25] is interpreted as the aversion an investor has toward uncertain parameter estimates, particularly in expected returns. The authors introduce a new optimization problem that accounts for ambiguity aversion. In addition to the standard mean-variance criterion for asset allocation, two new elements are included: a constraint on the estimated expected return to lie within a confidence interval, reflecting the possibility of estimation error, and the maximization of the mean-variance criterion. The optimization problem is formulated as a max–min problem, balancing the trade-off between maximizing expected utility and minimizing ambiguity. Two representations about the uncertainty of expected returns are provided: separate representation and joint representation. In separate representation, ambiguity aversion is assumed to be common across all assets, while ambiguity itself is considered asset-specific. This approach enables the expression of a probabilistic statement regarding the probability of all returns falling within the given confidence intervals. Joint representation, on the other hand, considers the non-independence of returns and formulates the confidence intervals for all assets jointly. The optimization problem is solved analytically under both representations. The results demonstrate a shrinkage behavior in asset allocation towards the minimum-variance portfolio as the ambiguity aversion parameter increases. The shrinkage factor is expressed as a combination of the weight vectors of the minimum-variance and mean-variance portfolios. An additional extension is proposed, considering the investor's belief in the Capital Asset Pricing Model (CAPM). The uncertainty aversion is tied to the degree of divergence of the regression constant ($\alpha$) from zero. This introduces a sensitivity parameter ($\kappa$) that allows the model to account for the investor's varying trust in historical data over time.

## 5. Methodology

### 5.1. Hypothesis

To improve the understanding of portfolio optimization and its effectiveness in dynamic financial markets, I established a number of hypotheses for this research. First, I predicted that models that include parameter uncertainty may perform better than the original tangency portfolio, which was static. Due to the understanding that financial markets are dynamic and always changing, models that incorporate parameter uncertainty should perform better by being able to adapt more readily to market swings. Second, I predicted that utility-specific portfolios would perform better than benchmark allocations. This expectation stemmed from the notion that portfolios customized to the preferences, risk tolerance, and goals of individual investors are better suited to meet specific requirements than general benchmark allocations and may even produce more advantageous results. Longer estimation periods, which contribute to larger datasets for parameter estimation, may yield more accurate parameter estimations, lowering uncertainty and enhancing portfolio performance overall, according to the research. The reasoning behind this is based on the idea that a bigger sample size makes it easier to grasp market dynamics precisely, which strengthens the portfolio optimization process. Furthermore, I proposed that the out-of-sample analysis would need to exhibit coherence with anticipated outcomes from the in-sample research, thus bolstering the credibility and dependability of the suggested model. Lastly, I investigated the theory that, in an uncertain environment, portfolios with short-sale constraints might perform better. The premise behind this notion is that restrictions on short sales can act as a risk management tool, limiting the potential for uncontrollable negative outcomes linked to short positions. Better risk-adjusted performance may result from a more cautious approach under uncertain market conditions, made easier by short-sale limitations. Collectively, these theories advanced a thorough examination of how parameter uncertainty, utility-driven portfolio design, length of the estimating

period, and short-sale constraints affect the performance of optimal portfolios in volatile and dynamic financial markets. Each hypothesis is supported by logical reasoning that serves as a solid basis for empirical testing and analysis with the goal of expanding the knowledge of efficient portfolio optimization techniques in a range of market scenarios.

### 5.2. Benchmark Portfolios

The investment portfolio evaluation involves considering benchmark allocations that were not optimized in a utility setting. These benchmarks include the minimum-variance allocation, $1/N$, risk parity, and lower partial moments allocation. In this section, the $1/N$ and risk parity allocations are established as benchmark strategies. The equal weight strategy, also known as the Talmudian investment strategy, is introduced as the first benchmark strategy outside the Markowitz setting. The motivation for including the equal weight strategy as a benchmark comes from research by [13,29]. Ref. [13] compared the naïve equal-weight ($1/N$) asset allocation strategy with various extensions of the classic mean-variance setting, using it as a benchmark to evaluate optimized portfolios. Ref. [29] found that the $1/N$ portfolio is not only a hypothetical application but is also applied in practice, as some individuals evenly spread their contributions among assets without considering historical measures for risk or return. The $1/N$ portfolio allocates equal weights to each asset, with the portfolio weights determined by the formula $\pi_i = 1/N$ for all $i$, where $N$ is the number of assets. As it does not consider historical measures for risk or return, its performance depends on the choice of assets and their correlation structure. Risk parity, or equally weighted risk contribution allocation, is introduced as a benchmark strategy that focuses solely on the risk associated with adding an asset to the portfolio, without considering expected returns. The portfolio weights are determined to ensure that the risk contribution of each asset is equal. The concepts of Marginal Risk Contribution (MRC) and Total Risk Contribution (TRC) are defined, where MRC represents the marginal risk added to the portfolio risk when increasing the weight of an asset, and TRC shows the contribution of an asset towards the portfolio risk. To achieve an equally weighted risk contribution portfolio, the TRC of each asset must be equal. This condition is expressed as $\pi_i \frac{\partial \sigma p}{\partial \pi i} = \pi_J \frac{\partial \sigma p}{\partial \pi j} = \lambda$ for all $i$ and $j$, where $\lambda$ is a constant depending on the final allocation of weights. The optimization problem for risk parity involves minimizing the squared difference in TRC between assets subject to the "sum-to-one" weight constraint. Before optimization, the final composition of the portfolio is unknown, and the total portfolio risk is also unknown. Unlike the closed-form solution in the mean-variance setting, the risk parity solution usually requires numerical optimization methods. The optimization problem is expressed as the minimization of the squared difference in TRC between assets subject to the "sum-to-one" weight constraint.

### 5.3. Short-Sale Constraint

In the past, there were no restrictions for selling high-risk investments short. Not every investor, though, has access to this approach. Furthermore, depending on the input parameters, the model can produce substantial negative positions that are not feasible. Therefore, it is interesting to look at the expected performance in such constrained situations. We will examine the restricted scenario in both studies, wherein an extra requirement is placed: every portfolio weight ($\pi_i$) needs to be non-negative, denoted by $\pi_i \geq 0$ for every $i$. With the exception of the $1/N$ strategy, none of the optimization problems for each approach have closed-form solutions; hence, numerical methods are needed to solve them. Because it is difficult to apply under constraints, ref. [3]'s three-fund strategy is not included in the analyses that forbid short sales.

## 6. Preliminary Analysis

### 6.1. Dataset Description

The Monte Carlo simulation and out-of-sample performance analysis were conducted using a set of asset returns based on the selection of [25], allowing for a comparison of

results. In addition to the assets chosen by [25], an index for the market was included, extending the time period of return series from July 2001 to February 2018, resulting in 577 months of data instead of 379, as shown in Table 7. To enable comparisons, all indices were quoted in the local currency, and the values were converted to USD using exchange rates on the day of the return observation. A risk-free rate, representing an investment with no risk of bankruptcy and no reinvestment risk, was chosen based on the 3-month US T-Bill. The market rate serves as a proxy for the risk-free rate during the period from 1 January 1970 to 1 February 2018, and its fluctuation is highlighted over the 48 years, as shown in Figure 5. The choice of return frequency for the Monte Carlo simulation and out-of-sample analysis is crucial. Monthly returns were selected for comparability with previous studies, including [3,13,25,26]. The discussion also touches on the frequency considerations, emphasizing that the chosen frequency should align with the investment strategy and readjustment frequency. The estimation of beta coefficients and other parameters involves selecting an appropriate estimation window length. Ref. [30] and others acknowledge the complexity of this choice, as too short a period may lead to biased estimates due to extreme values, while too long a period may not reflect changing asset risk profiles. Scholars such as [1] recommend a time frame of 48 to 84 months, while [25] use an estimation window of 120 months. This analysis examineddifferent window lengths from 48 to 120 months to assess their impact on expected and out-of-sample performance. Summary statistics of the return series, including mean returns, skewness, and kurtosis, revealed variations among indices, as shown in Table 8. Notably, skewness is negative for most indices, indicating more frequent small positive movements and less frequent large downward movements. The kurtosis values suggest fatter tails in the return distributions, with extreme returns more likely than implied by a normal distribution. A Shapiro–Wilk test for normality was performed, as shown in Table 9, and the results indicate that the assumption of normally distributed returns does not hold for all single-index return series or the combined dataset with a high confidence level.

**Table 7.** Indices considered for analysis.

| Countries | Ticker | Abbreviation | Currency |
| --- | --- | --- | --- |
| Italy | MSITALL | ITA | EUR (ITL) |
| Germany | MSGERML | GER | EUR (DEM) |
| Switzerland | MSSWITL | SWI | CHF |
| Canada | MSCNDAL | CAN | CAD |
| USA | MSUSAML | USA | USD |
| UK | MSUTDKL | UK | GBP |
| France | MSFRNCL | FRA | EUR (FF) |
| Japan | MSJPANL | JPN | JPY |
| World | MSWRLD | WRLD | USD |

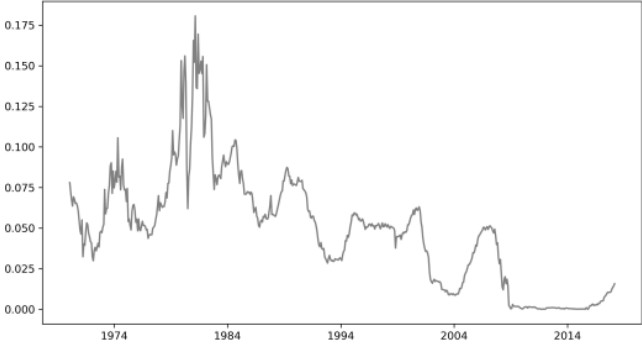

**Figure 5.** Annualized risk-free yield.

**Table 8.** Summary statistics of return series data used.

| Asset | $\hat{\mu}$ | $\hat{\sigma}$ | Min | Max | Skew | Kurtosis |
|---|---|---|---|---|---|---|
| ITA | 0.191% | 7.53% | −52.91% | 26.39% | −0.68 | 4.57 |
| GER | 0.542% | 6.43% | −24.92% | 29.67% | −0.44 | 2.02 |
| SWI | 0.685% | 5.28% | −19.14% | 23.31% | −0.36 | 1.73 |
| CAN | 0.482% | 5.73% | −30.05% | 20.72% | −0.78 | 3.19 |
| USA | 0.553% | 4.47% | −24.36% | 16.17% | −0.71 | 2.95 |
| UK | 0.429% | 6.06% | −24.09% | 44.01% | 0.35 | 5.47 |
| FRA | 0.513% | 6.54% | −31.35% | 22.05% | −0.55 | 1.83 |
| JPN | 0.610% | 6.12% | −28.11% | 22.20% | −0.02 | 1.28 |
| WRLD | 0.521% | 4.37% | −20.84% | 14.14% | −0.70 | 2.29 |

**Table 9.** Shapiro–Wilk test statistics and *p*-values; main dataset.

| Asset | $W$ | $p$-Value | Reject |
|---|---|---|---|
| ITA | 0.96 | 1.3E-10 | YES |
| GER | 0.97 | 2.8E-09 | YES |
| SWI | 0.97 | 6.7E-09 | YES |
| CAN | 0.96 | 4.2E-12 | YES |
| USA | 0.96 | 5.8E-11 | YES |
| UK | 0.95 | 1.2E-12 | YES |
| FRA | 0.98 | 4.7E-08 | YES |
| JPN | 0.99 | 2.8E-04 | YES |
| WRLD | 0.9625 | 5.4E-11 | YES |
| All Assets | 0.9661 | 2.5E-33 | YES |

*6.2. Empirical Analysis of Expected Certainty Equivalent Return (CER)*

The analysis of expected performance in portfolio allocations involves the consideration of investor preferences represented by a utility function. Each asset allocation strategy results in an uncertain future wealth value, and the level of expected utility serves as a measure of preference. However, utility functions are of ordinal character, making direct comparisons in cardinal terms challenging. To address this, the focus was here placed on the CER, which represents the risk-free return making an investor indifferent to investing in the risky alternative, allowing for cross-comparability among different levels of risk aversion. A Monte Carlo simulation was employed to incorporate randomness into the model and determine statistical values where analytical solutions may have been unavailable. The simulation involved choosing a set of expected return, variance, and covariance parameters for the eight assets and the market, as shown in Table 10. These parameters were calculated as sample mean and variance–covariance estimators from the previously introduced assets. The CER was then calculated based on these parameters. The optimal asset allocation maximizing the utility of a CARA investor was determined, consisting of an investment in the tangency portfolio and the risk-free asset. Returns for all assets and the market were simulated using a multivariate normal distribution, covering various time spans corresponding to the desired estimation window length. Portfolio weights for each asset allocation were determined, considering the optimal weight in the risky assets when a risk-free asset was included. Using the hypothetical knowledge of certain parameters, the expected portfolio return and standard deviation were calculated for each strategy. The CER was then calculated for the optimal asset allocation based on true return parameters, as well as for all portfolios with expected return and standard deviation, as defined above. The

CER was derived in the CARA utility setting, representing the risk-free wealth for which the investor is indifferent between investing in the asset offering a terminal wealth equal to the CE or the risky allocation. To capture the uncertainty induced by estimation, the process was repeated 10,000 times in the Monte Carlo simulation. The mean of the expected CER results from the simulations' sample was calculated, allowing for the evaluation of the expected performance of an allocation under uncertainty.

**Table 10.** $\Sigma^*$ and $\mu^*$ annualized figures.

| Asset | ITA | GER | SWI | CAN | USA | UK | FRA | JPN | WRLD |
|-------|-----|-----|-----|-----|-----|-----|-----|-----|------|
| **ITA** | 6.81% | 3.03% | 2.36% | 2.32% | 1.79% | 2.71% | 3.60% | 2.02% | 2.29% |
| **GER** | 3.03% | 4.96% | 2.96% | 2.41% | 2.04% | 2.70% | 3.72% | 1.99% | 2.45% |
| **SWI** | 2.36% | 2.96% | 3.35% | 2.01% | 1.75% | 2.44% | 2.88% | 1.85% | 2.07% |
| **CAN** | 2.32% | 2.41% | 2.01% | 3.94% | 2.30% | 2.52% | 2.56% | 1.59% | 2.32% |
| **USA** | 1.79% | 2.04% | 1.75% | 2.30% | 2.40% | 2.10% | 2.10% | 1.21% | 2.08% |
| **UK** | 2.71% | 2.70% | 2.44% | 2.52% | 2.10% | 4.40% | 3.07% | 1.92% | 2.44% |
| **FRA** | 3.60% | 3.72% | 2.88% | 2.56% | 2.10% | 3.07% | 5.14% | 2.13% | 2.54% |
| **JPN** | 2.02% | 1.99% | 1.85% | 1.59% | 1.21% | 1.92% | 2.13% | 4.50% | 2.14% |
| **WRLD** | 2.29% | 2.45% | 2.07% | 2.32% | 2.08% | 2.44% | 2.54% | 2.14% | 2.29% |
| $\mu^*$ | 2.30% | 6.50% | 8.22% | 5.79% | 6.64% | 5.14% | 6.15% | 7.32% | 6.25% |

### 6.3. Empirical Analysis of Out-of-Sample Performance

Out-of-sample performance evaluation involves actively trading the previously proposed investment strategies using the same allocations. In this case, the estimation of parameters was conducted within chosen windows, and return data from those windows was utilized to estimate input parameters for calculating portfolio weights. The same estimation window lengths (48, 60, 72, 96, 120 months) considered in the previous analysis were applied. The trading period spanned from January 1980 to January 2018, and portfolios were adjusted on a monthly basis. The updated portfolio weights were determined based on the estimation window ahead of the readjustment date until the strategy was implemented for the last time and sold at $t = E$, as shown in Figure 6. The expected performance was initially assessed using the CER, calculated based on the mean and standard deviation of returns resulting from the implementation of each strategy. This measure provides insight into the average performance of each strategy. Additional performance measures were introduced for a comprehensive evaluation:

1.   Sharpe Ratio (SR): The Sharpe ratio is a common tool for evaluating risk-adjusted return. It assesses whether returns are generated due to excessive risk or smart investment decisions. The excess returns of asset $i$ in period $t\left(r_{e_{i,t}}\right)$ are calculated as the difference between the asset return $r_{i,t}$ and the risk-free rate $r_{f_t}$. The Sharpe ratio for asset $i(SR_i)$ is then the ratio of the mean excess return to the standard deviation of excess returns.

2.   Turnover (TO): Turnover measures the number of assets that need to be traded on average while implementing a trading strategy. It includes all rebalancing made to the portfolio over the considered period. The TO is calculated as the average sum of the absolute values of trades across all assets. It serves as an indicator for trading costs, which are proportional to the wealth traded.

These performance measures provide a comprehensive assessment of the strategies' out-of-sample performance. The Sharpe ratio indicates risk-adjusted returns, allowing for comparisons between strategies, while turnover reflects the trading activity and associated costs. These measures collectively contribute to a thorough understanding of the strategies' effectiveness and efficiency in real-world trading scenarios.

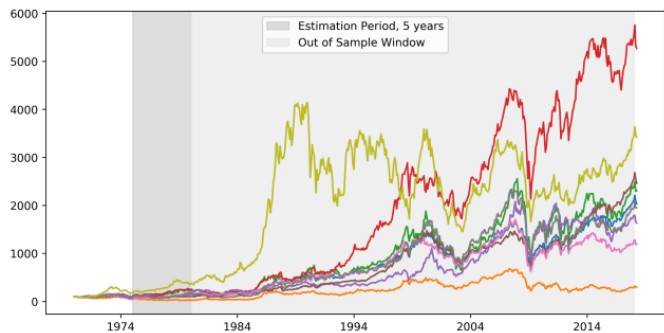

**Figure 6.** Illustration of first estimation window and out-of-sample window.

## 7. Analysis Results

The results of the two empirical analyses are presented below. Both analyses considered scenarios with and without a short-sale restriction on risky assets, as well as a risk aversion of $\gamma = 1$. In addition, we compared the CER across different $\gamma$ values (with $T$ held constant) and made use of a control dataset. This comprehensive approach enabled us to verify the robustness and strength of our findings. Firstly, we evaluated the expected performance in terms of CER. Secondly, we turned our attention to assessing the out-of-sample performance of different strategies.

### 7.1. Expected CER Analysis

The results of the Monte Carlo analysis evaluating the expected certainty equivalent return are shown below.

#### 7.1.1. Expected CER When Short Sales Are Allowed

The portfolio weights of the certain allocation, allowing for short sales, imply an expected monthly return of 1.32% and a monthly portfolio standard deviation of 9.64%, resulting in a certain CER of 0.855%. Since this certain allocation maximizes expected utility, any other allocation will imply a lower CER. The Monte Carlo analysis, which allowed for short sales, yielded the results depicted in Table 11 and Figure 7. The analysis aimed to test several hypotheses:

- Hypothesis 1: Models that account for parameter uncertainty outperform the original tangency portfolio. The results support this hypothesis as the tangency portfolio performs notably poorer under uncertainty (ranging between −14.60% and −3.33% depending on the observation length). In contrast, the LW-tangency, GUW, and the three-fund allocation outperform the tangency allocation under uncertainty. The GUW allocation, in particular, shows a significant improvement, even though its intention is not to optimize performance. The positive effect decreases with increasing estimation window length.
- Hypothesis 2: Utility-optimized portfolios perform better relative to benchmark allocations. Contradictory findings emerged regarding this hypothesis. Among the utility-optimized models, GUW and the three-fund portfolios show the best performance, while the tangency, LW-tangency, and mean-variance allocation perform more modestly. Benchmark portfolios, including the minimum-variance portfolio, show positive expected CER, with the minimum-variance portfolio comparable to the GUW allocations.
- Hypothesis 3: Larger estimation windows lead to more-accurate parameter estimates (less uncertainty) and therefore better performance. This pattern is confirmed for all utility-optimized portfolios and the minimum-variance portfolio. However, this trend is not observed for other portfolios, such as in the Lower Partial Moments (LPMs) setting, where longer estimation windows do not consistently impact performance positively. The positive effect of longer estimation windows on performance improvement decreases as the estimation window length increases. This converging behavior

suggests that further increasing the estimation length has a decreasing positive impact on expected performance improvement.

**Table 11.** Expected monthly CER $\gamma = 1$; short sales allowed.

| | Estimation Window Length $T$ | | | | |
|---|---|---|---|---|---|
| **Utility-Optimized Portfolios** | **48** | **60** | **72** | **96** | **120** |
| **Certain Tangency** | 0.855% | 0.855% | 0.855% | 0.855% | 0.855% |
| **Tangency** | −14.603% | −9.856% | −7.274% | −4.690% | −3.334% |
| **LW-Tangency** | −12.776% | −9.056% | −6.835% | −4.540% | −3.271% |
| **Mean-Variance** | −11.542% | −8.047% | −5.912% | −3.863% | −2.691% |
| **GUW $\varepsilon$ = 0.5** | −0.966% | −0.528% | −0.233% | 0.016% | 0.016% |
| **GUW $\varepsilon$ = 1** | 0.137% | 0.251% | 0.336% | 0.401% | 0.443% |
| **GUW $\varepsilon$ = 1.5** | 0.397% | 0.434% | 0.466% | 0.491% | 0.508% |
| **GUW $\varepsilon$ = 2** | 0.462% | 0.482% | 0.500% | 0.515% | 0.526% |
| **GUW $\varepsilon$ = 3** | 0.493% | 0.504% | 0.517% | 0.526% | 0.534% |
| **GUW ext.** | 0.433% | 0.438% | 0.452% | 0.457% | 0.466% |
| **TF—Kan Zhou** | −1.016% | −0.704% | −0.475% | −0.227% | −0.064% |
| **Benchmark Portfolios** | | | | | |
| **Min-Variance** | 0.492% | 0.496% | 0.500% | 0.501% | 0.502% |
| **1/N** | 0.393% | 0.393% | 0.393% | 0.393% | 0.393% |
| **RP** | 0.408% | 0.408% | 0.409% | 0.409% | 0.409% |
| **LPM (6.2%)** | 0.436% | 0.434% | 0.430% | 0.430% | 0.402% |

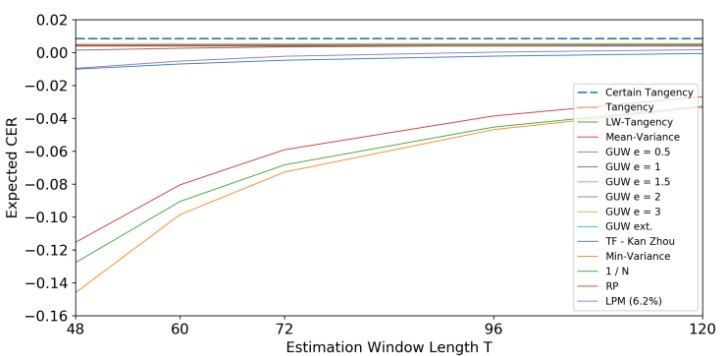

**Figure 7.** Expected CER; $\gamma = 1$; short sales allowed.

### 7.1.2. Expected CER When Short Sales Are Constrained

The introduction of additional constraints on the optimal asset allocation has a notable impact on the certain CER. When a short-sale constraint is included, the optimal asset allocation changes considerably. In the unconstrained case, negative weights on risky assets were allowed, but with the short-sale constraint, these assets were excluded from the allocation. The expected return decreases to 0.72%, the portfolio standard deviation decreases to 5.75%, and the resulting CER is reduced to 0.555%. Performing the same analysis as before with the short-sale constraint yielded the results depicted in Table 12 and Figure 8. The analysis aimed to test several hypotheses:

**Table 12.** Expected monthly CER $\gamma = 1$; no short sales.

| | Estimation Window Length $T$ | | | | |
|---|---|---|---|---|---|
| **Utility-Optimized Portfolios** | **48** | **60** | **72** | **96** | **120** |
| **Certain Tangency** | 0.555% | 0.555% | 0.555% | 0.555% | 0.555% |
| **Tangency** | −1.670% | −1.132% | −0.865% | −0.466% | −0.271% |
| **LW-Tangency** | −1.652% | −1.126% | −0.864% | −0.467% | −0.271% |
| **Mean-Variance** | 0.361% | 0.374% | 0.381% | 0.397% | 0.410% |
| **GUW $\varepsilon = 0.5$** | 0.5 0.444% | 0.448% | 0.452% | 0.457% | 0.462% |
| **GUW $\varepsilon = 1$** | 1 0.457% | 0.460% | 0.463% | 0.469% | 0.473% |
| **GUW $\varepsilon = 1.5$** | 1.5 0.463% | 0.466% | 0.470% | 0.474% | 0.478% |
| **GUW $\varepsilon = 2$** | 2 0.466% | 0.470% | 0.473% | 0.478% | 0.482% |
| **GUW $\varepsilon = 3$** | 3 0.471% | 0.474% | 0.478% | 0.482% | 0.485% |
| **GUW ext.** | 0.462% | 0.464% | 0.466% | 0.469% | 0.472% |
| **TF—Kan Zhou** | - | - | - | - | - |
| **Benchmark Portfolios** | | | | | |
| **Min-Variance** | 0.474% | 0.478% | 0.481% | 0.485% | 0.487% |
| **1/$N$** | 0.393% | 0.393% | 0.393% | 0.393% | 0.393% |
| **RP** | 0.408% | 0.408% | 0.408% | 0.408% | 0.408% |
| **LPM (6.2%)** | 0.372% | 0.362% | 0.346% | 0.323% | 0.306% |

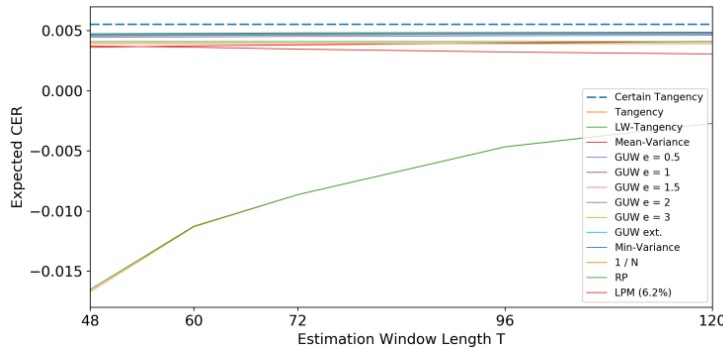

**Figure 8.** Expected CER; $\gamma = 1$; short sales not allowed.

- Hypothesis 1: Models that account for parameter uncertainty outperform the original tangency portfolio. The GUW allocations still clearly outperform the tangency allocation for all levels of epsilon, although the advantage has decreased notably compared to the unconstrained case. The positive impact of the shrunk covariance matrix fades out due to the constraint, leading to the tangency and LW-tangency performing equally well.

- Hypothesis 2: Utility-optimized portfolios perform better relative to benchmark allocations. Contrary to the hypothesis, there is no clear evidence of better performance for utility-optimized allocations. The minimum-variance portfolio, which is not dependent on $\gamma$, shows good performance, challenging the expectation that utility-optimized portfolios should outperform.

- Hypothesis 3: Larger estimation windows lead to more accurate parameter estimates (less uncertainty) and therefore better performance. Short-sale-constrained portfolios exhibit similar behavior regarding the length of the estimation window as in the non-constrained case. Utility-optimized portfolios and the minimum-variance portfolio perform better with longer estimation windows.

- Hypothesis 4: Short-sale-constrained portfolios perform better under uncertainty than non-constrained asset allocations. The short-sale constraint has a dual impact. It improves the expected performance of the original tangency and mean-variance allocations, while portfolios considering uncertainty and the minimum-variance portfolio perform slightly worse.

### 7.1.3. Different Values of $\gamma$

The analysis of different values of risk aversion ($\gamma$) provides valuable insights into how investor preferences impact the performance of various asset allocations under uncertainty. The results, considering a fixed estimation window length of $T = 60$, are summarized in Table 13. Several observations can be made from the findings:

**Table 13.** Expected monthly CER varying $\gamma$; $T = 60$; short sales allowed.

| | Risk Aversion Coefficient$\gamma$ | | | | |
|---|---|---|---|---|---|
| **Utility-Optimized Portfolios** | **0.5** | **1.0** | **1.5** | **2.0** | **3.0** |
| **Certain Tangency** | 1.320% | 0.855% | 0.700% | 0.622% | 0.545% |
| **Tangency** | −20.102% | −9.856% | −6.441% | −4.733% | −2.165% |
| **LW-Tangency** | −18.501% | −9.056% | −5.907% | −4.333% | −2.019% |
| **Mean-Variance** | −16.552% | −8.047% | −5.247% | −3.871% | −1.823% |
| **GUW $\varepsilon$ = 0.5** | −1.339% | −0.528% | −0.307% | −0.230% | −0.074% |
| **GUW $\varepsilon$= 1** | 0.203% | 0.251% | 0.219% | 0.174% | 0.136% |
| **GUW $\varepsilon$ = 1.5** | 0.518% | 0.434% | 0.361% | 0.295% | 0.213% |
| **GUW $\varepsilon$ = 2** | 0.564% | 0.482% | 0.409% | 0.343% | 0.249% |
| **GUW $\varepsilon$ = 3** | 0.573% | 0.504% | 0.440% | 0.378% | 0.281% |
| **GUW ext.** | 0.533% | 0.438% | 0.360% | 0.291% | 0.192% |
| **TF—Kan Zhou** | −1.798% | −0.704% | −0.340% | −0.158% | 0.025% |
| **Benchmark Portfolios** | | | | | |
| **Min-Variance** | 0.543% | 0.496% | 0.449% | 0.402% | 0.315% |
| **1/N** | 0.448% | 0.393% | 0.339% | 0.284% | 0.175% |
| **RP** | 0.461% | 0.408% | 0.356% | 0.304% | 0.199% |
| **LPM (6.2%)** | 0.500% | 0.434% | 0.367% | 0.300% | 0.166% |

- Impact of risk aversion on tangency and mean-variance allocations: As risk aversion increases, the expected performance of the tangency and mean-variance allocations under uncertainty improves. This suggests that investors with higher risk aversion find these strategies more appealing. The expected performance for highly risk-averse investors is less dispersed compared to less risk-averse investors, indicating more consistency in outcomes for risk-averse individuals.
- Divergent patterns among asset allocations considering uncertainty: There is no common pattern among asset allocations when considering uncertainty. Different strategies exhibit varied behavior with changing risk aversion. The LW-tangency and three-fund allocation show an increasing expected CER with higher risk aversion ($\gamma$), suggesting that these strategies may be more suitable for risk-averse investors. The behavior of the GUW allocation is more nuanced, showing increasing expected CER for lower values of ambiguity aversion ($\varepsilon$ = 0.5) but a reversal of this trend for higher values of $\varepsilon$.
- Performance of benchmark portfolios: Interestingly, the minimum-variance allocation stands out as the best performer for risk aversion levels from $\gamma$ = 1.5 onward, even

though it decreases in performance as risk aversion decreases. This contradicts the trend observed in other benchmark portfolios.

These findings underscore the importance of considering investor-specific characteristics, such as risk aversion, when evaluating portfolio performance under uncertainty. Different investors may prefer different strategies based on their risk preferences, and there is no one-size-fits-all solution. The nuanced patterns observed highlight the complexity of decision making in portfolio construction and the need for a tailored approach that aligns with individual investor characteristics.

### 7.1.4. Expected CER Using Control Data

The control dataset, which utilized a risk aversion parameter of $\gamma = 1$ and allowedfor short sales, provides additional insights into the performance of various asset allocation strategies, as shown in Table 13. The findings can be summarized based on the hypotheses outlined:

- Hypothesis 1: The LW-tangency allocation demonstrates superior expected performance compared to the tangency allocation under uncertainty. This aligns with the hypothesis and reinforces the idea that incorporating parameter uncertainty can lead to improved outcomes. Similarly, the GUW and the three-fund allocation outperform the tangency portfolio, supporting the notion that considering uncertainty in parameter estimates enhances the performance of these portfolios.
- Hypothesis 2: Contrary to the hypothesis, utility-optimized portfolios do not consistently outperform benchmark allocations in the control dataset. Among the utility-optimized portfolios, only those considering uncertainty show strong performance. Benchmark portfolios perform well in the control dataset, challenging the idea that utility-optimized strategies consistently lead to better outcomes.
- Hypothesis 3: The control dataset exhibits a similar pattern to the main dataset, supporting the hypothesis. Utility-optimized portfolios and the minimum-variance allocation benefit from larger estimation windows, leading to more accurate parameter estimates and better performance. The relationship does not hold for portfolios following $1/N$, risk parity, and LPM, indicating that larger estimation windows may not uniformly improve the performance of all allocation strategies.
- Hypothesis 4: Consistent with the main dataset, short-sale-constrained portfolios in the control dataset exhibit less dispersed results. Portfolios that performed modestly in the unconstrained case now show improved performance here, where short sales were restricted. The constrained portfolios, especially those that performed modestly without constraints, demonstrate a more consistent and favorable performance under uncertainty.

## 8. Discussion

While exploring several asset allocation strategies, this paper acknowledges the inherent assumptions and limitations of these models. The correct estimation of differences, connections, and predicted returns is critical to the precision of asset division. Notably, the Markowitz framework serves as an example of the difficulty of parameter uncertainty, which adds a degree of unpredictability that may affect actual outcomes. The manner in which market behavior, relationships, and the stability of risk–return profiles are assumed throughout analysis also greatly influence the final results. It is crucial to recognize possible deviations from presumptive market circumstances when applying these conclusions to actual situations. The study makes the assumption that markets behave rationally and uses historical data to assess performance; yet, unanticipated market shocks, changes in investor mood, or disruptions in the economy could put the viability of the suggested methods in jeopardy. As such, the study recognizes the dynamic nature of financial markets and argues for a careful interpretation of the data. Investors and portfolio managers need to be aware of these constraints in real-world applications. Even while the study's conclusions can offer helpful advice, it is important to take individual investment goals, risk tolerance,

and the state of the market into account. The study also highlights the significance of a careful evaluation of the effects of short-sale limitations on assets, which is a point that is frequently disregarded in practice.

### 9. Conclusions

In conclusion, this paper explores the difficulties related to uncertainty in the Markowitz framework's parameters, recognizing that assuming certainty in input parameters is not realistic. By thoroughly examining different methods, including Bayesian estimation with shrinkage techniques and an ambiguity aversion model, the study compares their performance with alternative strategies. The results show that considering uncertainty improves the expected performance of the Markowitz model. Specifically, using Ledoit–Wolf shrinkage consistently enhances both expected and real-life out-of-sample performance in a tangency setting. However, the paper points out that relying on expected performance as a predictor in tangency and mean-variance allocation scenarios may not be reliable. Moreover, the proposed expansion of the ambiguity aversion framework, taking into account how dependable the data arefor estimating parameters, shows promising outcomes, even with reduced shrinkage towards minimum-variance allocation. Notably, the research challenges the notion that extending the estimation period consistently enhances out-of-sample performance. Despite having competitive benchmark portfolios, the paper expresses valid concerns regarding the practicality and workability of the Markowitz framework and its extensions.

**Funding:** This research received no external funding.

**Institutional Review Board Statement:** Not applicable.

**Informed Consent Statement:** Not applicable.

**Data Availability Statement:** The data will be available upon reasonable request.

**Conflicts of Interest:** The author declares no conflicts of interest.

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
