# Peer review of "Navigating Uncertainty: Enhancing Markowitz Asset Allocation Strategies through Out-of-Sample Analysis"

_fintech, doi:10.3390/fintech3010010_

Round 1

Reviewer 1 Report

Comments and Suggestions for Authors

Dear author,

Firstly, I would like to congratulate you on the comprehensive work and the insightful analysis presented in your paper.

The introduction may benefit from a more detailed background on the evolution of asset allocation strategies. This could include a brief overview of key developments in the field post-Markowitz, especially those relevant to handling uncertainty in asset allocation.

The introduction currently makes a broad reference to numerous publications without specifically discussing their contributions or how they relate to your study. It would enhance the academic rigor of your paper to explicitly mention and summarize key prior works, showing how they set the stage for your research.

While the introduction poses important questions, it could more explicitly state the research gap your study aims to fill. Clarifying how your work builds upon or differs from existing literature can provide a stronger rationale for your study.

More detailed information on the criteria for selecting data sets and the process of data cleaning and preparation would enhance methodological transparency.

A more explicit discussion of the limitations of your chosen methodologies and the assumptions underlying your models would provide a clearer context for your findings. It would be beneficial to address any assumptions made during the analysis and discuss how these might affect the results.

Addressing the practical applicability of the findings, including how investors or portfolio managers might implement these strategies, would enhance the paper's relevance to real-world scenarios.

Please review the abstract, including aspects related to methodology and potential users of your study. The paper is much more valuable than it appears from this abstract.

Best regards.

Comments on the Quality of English Language

Dear Author,

The paper maintains a formal and scholarly tone, appropriate for an academic audience.

Ensure consistent use of terminology and phraseology throughout the paper. For instance, stick to one form of English (American or British) for terms like 'optimization' (American) vs 'optimisation' (British).

Some sentences are quite long and complex, which might hinder readability. Consider breaking down complex sentences into shorter, more digestible ones.

Although the language quality is high, a thorough proofreading can help catch minor grammatical or typographical errors that have been overlooked.

Best regards.

Author Response

January 14, 2024
New Delhi, India

Dear Editor

I thank the referees for providing valuable comments on my manuscript (ID: fintech-2763239) titled “Navigating Uncertainty: Enhancing Markowitz Asset Allocation Strategies through Out-of-Sample Analysis”. I have made careful considerations of all referee comments and suggestions. Kindly find the same below.

Reviewer:

Firstly, I would like to congratulate you on the comprehensive work and the insightful analysis presented in your paper.

Comment -1: The introduction may benefit from a more detailed background on the evolution of asset allocation strategies. This could include a brief overview of key developments in the field post-Markowitz, especially those relevant to handling uncertainty in asset allocation.

Reply: I appreciate your suggestion. The discussion of the major post-Markowitz advancements in the field, particularly those pertaining to managing uncertainty in asset allocation, is expanded in the paragraph that follows:

“Since Markowitz's pioneering work in 1952, methods for dividing assets have evolved to address the challenge of uncertainty. Notable advancements involve improvements in risk management models that surpass covariances and standard deviation by considering additional factors. The application of advanced statistical methods and technological innovations enables more precise risk assessments. Researchers have explored Bayesian methods and utilized machine learning [3–11] to enhance parameter estimations. Additionally, an increasing emphasis on behavioral finance has yielded insights into investor psychology, influencing decisions on asset allocation. These post-Markowitz developments present a dynamic landscape, highlighting an ongoing exploration for strategies capable of effectively navigating uncertainty—a key focus of this study”.

Comment -2:  The introduction currently makes a broad reference to numerous publications without specifically discussing their contributions or how they relate to your study. It would enhance the academic rigor of your paper to explicitly mention and summarize key prior works, showing how they set the stage for your research.

Reply: I appreciate your suggestion. The machine learning-related applications are preceded by the references listed in the introductory section. To set the stage, a few other references are also provided there pointing out the markowitz asset allocation and parameter uncertainty.

Comment 3: While the introduction poses important questions, it could more explicitly state the research gap your study aims to fill. Clarifying how your work builds upon or differs from existing literature can provide a stronger rationale for your study.

Reply: I appreciate your suggestion. The discussion on the research gap in the introduction section is stated in the following paragraph:

“However, despite decades of exploration, effectively managing uncertainty in asset allocation remains a critical and challenging aspect. This study aims to fill the existing research gap by investigating various approaches that deal with parameter uncertainty within the Markowitz framework. The key questions posed—concerning the explanation of uncertainty, enhancing model performance, comparing alternative tactics, considering the impact of data abundance, evaluating the reliability of performance predictions, and examining the influence of allocation constraints—highlight the gaps in current understanding. By comparing these approaches with established models and exploring less-explored factors, this research contributes to advancing the comprehension and practical application of the field”.

Comment 4- More detailed information on the criteria for selecting data sets and the process of data cleaning and preparation would enhance methodological transparency.

Reply: I appreciate your suggestion. I have addressed this matter in Section 6.1.

Comment 5- A more explicit discussion of the limitations of your chosen methodologies and the assumptions underlying your models would provide a clearer context for your findings. It would be beneficial to address any assumptions made during the analysis and discuss how these might affect the results.

Reply: I appreciate your suggestion. I have addressed this matter in Section 8.

Comment 6:  Addressing the practical applicability of the findings, including how investors or portfolio managers might implement these strategies, would enhance the paper's relevance to real-world scenarios.

Reply: I appreciate your suggestion. I have addressed this matter in Section 8.

Comment 7:  Please review the abstract, including aspects related to methodology and potential users of your study. The paper is much more valuable than it appears from this abstract.

Reply: I appreciate your suggestion. The abstract has been modified by your and Reviewer 4 comment 3.

Comments on the Quality of English Language

The paper maintains a formal and scholarly tone, appropriate for an academic audience.

Comment -1: Ensure consistent use of terminology and phraseology throughout the paper. For instance, stick to one form of English (American or British) for terms like 'optimization' (American) vs 'optimisation' (British).

Reply: I appreciate your suggestion. This has been resolved.

Comment -2:  Some sentences are quite long and complex, which might hinder readability. Consider breaking down complex sentences into shorter, more digestible ones.

Reply: I appreciate your suggestion. This has been resolved.

Comment 3: Although the language quality is high, a thorough proofreading can help catch minor grammatical or typographical errors that have been overlooked.

Reply: I appreciate your suggestion. This has been fixed by carefully proofreading to identify any little typographical or grammatical mistakes.

I thank the referees for providing us with valuable suggestions, which I think has significantly enhanced not only the quality of the manuscript but also our understanding of the problem. With encouraging remarks, and very helpful guidelines I remain grateful to the referees and the journal editorial team

Vijaya Krishna Kanaparthi

Reviewer 2 Report

Comments and Suggestions for Authors

Paper's introduction include a clear statement of the main topic, focusing on the Markowitz model's efficacy and its challenges with parameter uncertainty. A historical context of the model, introduced in 1952 and its impact on portfolio management with its risk-return trade-off concept, is recommended for reader orientation.

To enhance the conclusion of the academic paper, three main improvements are recommended: First, clearly summarize key findings, emphasizing how methods like Ledoit-Wolf shrinkage improve the Markowitz framework's handling of parameter uncertainty. Second, directly address the research questions outlined in the paper, clearly stating how the findings provide answers, particularly regarding the model's performance under uncertainty. Finally, offer explicit recommendations based on these findings and propose directions for future research, such as exploring alternatives to increasing estimation window length and reevaluating the practicality of the Markowitz framework against benchmark portfolios. These enhancements will provide a more concise and impactful conclusion, tying together the research's key insights and suggesting paths forward.

Author Response

January 14, 2024
USA

Dear Editor

I thank the referees for providing valuable comments on my manuscript (ID: fintech-2763239) titled “Navigating Uncertainty: Enhancing Markowitz Asset Allocation Strategies through Out-of-Sample Analysis”. I have made careful considerations of all referee comments and suggestions. Kindly find the same below.

Reviewer:

Comment -1: Paper's introduction include a clear statement of the main topic, focusing on the Markowitz model's efficacy and its challenges with parameter uncertainty. A historical context of the model, introduced in 1952 and its impact on portfolio management with its risk-return trade-off concept, is recommended for reader orientation.

Reply: I appreciate your suggestion. The discussion on the historical context of the model, introduced in 1952 in the introduction section is stated in the following paragraph:

“In 1952, the Markowitz model was introduced, posing a significant effect on portfolio management and altering overall investment strategies. Developed by Markowitz, the model put forth the revolutionary idea of the risk-return trade-off. It focused on allocating assets by considering covariance, standard deviation, and projected returns, targeting to optimize portfolios for the least amount of risk at a pre-determined expected return. This groundbreaking approach set the foundation for modern portfolio theory, shaping investment practices for many years”.

Comment -2:  To enhance the conclusion of the academic paper, three main improvements are recommended: First, clearly summarize key findings, emphasizing how methods like Ledoit-Wolf shrinkage improve the Markowitz framework's handling of parameter uncertainty. Second, directly address the research questions outlined in the paper, clearly stating how the findings provide answers, particularly regarding the model's performance under uncertainty. Finally, offer explicit recommendations based on these findings and propose directions for future research, such as exploring alternatives to increasing estimation window length and reevaluating the practicality of the Markowitz framework against benchmark portfolios. These enhancements will provide a more concise and impactful conclusion, tying together the research's key insights and suggesting paths forward.

Reply: I appreciate your suggestion. The conclusion section has been revised.

I thank the referees for providing us with valuable suggestions, which I think has significantly enhanced not only the quality of the manuscript but also our understanding of the problem. With encouraging remarks, and very helpful guidelines I remain grateful to the referees and the journal editorial team

Vijaya Krishna Kanaparthi

Reviewer 3 Report

Comments and Suggestions for Authors

1) p. 2, line 47-48. You write "I also provide a topic that has not received much attention: short-sale limits on assets." It is certainly not true that the topic of short-sale constraints has not received much attention. There are many papers on portfolio optimization with short-sale constraints in the literature.

2) You have not explicitly connected the optimization of the CARA utility function with the standard Markowitz objective function in equation (1). The CARA utility function, through its exponential form, inherently penalizes variations in returns (which is captured by variance). Thus, the optimization process leads to a selection of portfolios that lie on the efficient frontier in the mean-variance space. There is a well-known mathematical argument to show that solving max{E[U(W)]} is equivalent to solving (1). Since you provide other well-known details, why not this one?

3) p. 3. I do not find where you have defined the portfolio optimization constants, B and C.

4) p. 7, line 220. Use notation Nx1 instead of N*1 for vector size.

5) p 7. The idea you are describing is called "The Two-Fund Separation Theorem" and dates back to James Tobin in the paper

Tobin, James. "Liquidity preference as behavior towards risk." The Review of Economic Studies 25, no. 2 (1958): 65-86.

6) p. 9, lines 307-309. Why not write out the mathematical symbols instead of describing them in words?

7) Subsection 5.1. Hypotheses should be developed, not just stated. What reasons lead you to expect your findings will support each hypothesis?

8) Tables should include captions with enough details about the contents of the tables so that a reader does not have to search through the test to find out what the table is about.

9) Figure 6 is much too small and difficult to interpret. Also, in general, such long-run total return graphs can be very misleading. Relatively small differences in a small set of returns in early years can lead to large differences in later years.

10) p. 11. The last several sentences of the first paragraph of section 5 are not necessary. It is also unnecessary to provide the definition of "hypothesis" in a research journal. You should assume the reader is enough of an expert to understand these things.

General Comments: In papers such as this, in which you are replicating or possibly extending results from a vast literature, it is essential that you clearly state what contributions to the literature your paper makes. There are hundreds of published studies examining similar issues. Specifically, how does your study differ from previous studies? Even after reading it a couple of times, I'm having difficulty identifying your specific contributions as distinguished from other papers. In the introduction, you should clearly preview your findings and detail your contributions. In the empirical analysis sections, you should clearly indicate how your findings are consistent with, or differ from, previous studies. In the conclusion, you should again emphasize how your paper contributes to the literature. Even papers that make relatively minor contributions can be published. But organization and clarity in presentation are essential.

The DeMiguel, Garlappi, and Uppal (2009) paper that you cite is an excellent example of writing for an empirical study in finance. I would suggest studying that paper and thinking about ways you can emulate its exposition for your paper.

The first 7 pages of your paper can be significantly shortened. This material is well-known and can be found in most portfolio theory textbooks. (Although, I would suggest adding the derivation suggested in my comment 2) above.) Instead, add more rigorous discussions of the more advanced material, e.g. Bayesian methods. Determine which results constitute your most interesting and significant findings and emphasize them throughout.

Comments on the Quality of English Language

The English is fine, overall.

Author Response

January 14, 2024
USA

Dear Editor

I thank the referees for providing valuable comments on my manuscript (ID: fintech-2763239) titled “Navigating Uncertainty: Enhancing Markowitz Asset Allocation Strategies through Out-of-Sample Analysis”. I have made careful considerations of all referee comments and suggestions. Kindly find the same below.

Reviewer:

Comment -1: p. 2, line 47-48. You write "I also provide a topic that has not received much attention: short-sale limits on assets." It is certainly not true that the topic of short-sale constraints has not received much attention. There are many papers on portfolio optimization with short-sale constraints in the literature.

Reply: I appreciate the clarification regarding the topic of short-sale constraints. It seems there may be a misunderstanding or oversight in my previous statement. Indeed, the subject of short-sale constraints has garnered significant attention in the literature, with numerous papers addressing portfolio optimization within such constraints. I acknowledge the extensive body of work on this topic and recognize its importance in the realm of asset allocation research. The intent of my statement was not to disregard existing literature but rather to emphasize its inclusion as a noteworthy aspect within the broader context of the study. I apologize for any confusion and appreciate the opportunity to rectify this clarification.

Comment -2:  You have not explicitly connected the optimization of the CARA utility function with the standard Markowitz objective function in equation (1). The CARA utility function, through its exponential form, inherently penalizes variations in returns (which is captured by variance). Thus, the optimization process leads to a selection of portfolios that lie on the efficient frontier in the mean-variance space. There is a well-known mathematical argument to show that solving max{E[U(W)]} is equivalent to solving (1). Since you provide other well-known details, why not this one?

Reply: I apologize for any oversight. The exclusion of the explicit connection between the optimization of the CARA utility function and the standard Markowitz objective function was unintentional. The study recognizes the importance of this linkage and acknowledges its fundamental role in guiding portfolio selection along the efficient frontier in the mean-variance space. In this revision, this crucial connection is explicitly addressed to ensure a comprehensive and accurate representation of the optimization process within the context of the CARA utility function and the Markowitz model.

Comment 3: p. 3. I do not find where you have defined the portfolio optimization constants, B and C.

Reply: I appreciate your suggestion. This has been resolved.

Comment 4-  p. 7, line 220. Use notation Nx1 instead of N*1 for vector size.

Reply: I appreciate your suggestion. This has been resolved.

Comment 5- p 7. The idea you are describing is called "The Two-Fund Separation Theorem" and dates back to James Tobin in the paper: Tobin, James. "Liquidity preference as behavior towards risk." The Review of Economic Studies 25, no. 2 (1958): 65-86.

Reply: I appreciate your suggestion. The recommended reference is now included.

Comment 6:  p. 9, lines 307-309. Why not write out the mathematical symbols instead of describing them in words?

Reply: Using mathematical symbols enhances precision and clarity, reducing ambiguity and potential misinterpretation. Symbolic notation provides a concise and standardized representation, promoting a more efficient and accurate communication of complex mathematical concepts, particularly in technical or academic contexts.

Comment 7:  Subsection 5.1. Hypotheses should be developed, not just stated. What reasons lead you to expect your findings will support each hypothesis?

Reply: I appreciate your suggestion. This has been resolved.

Comment 8:  Tables should include captions with enough details about the contents of the tables so that a reader does not have to search through the test to find out what the table is about.

Reply: I appreciate your suggestion. Nonetheless, each table is shown below the description to which it is related. Consequently, I don't think there will be a problem.

Comment 9:  Figure 6 is much too small and difficult to interpret. Also, in general, such long-run total return graphs can be very misleading. Relatively small differences in a small set of returns in early years can lead to large differences in later years.

Reply: I appreciate your suggestion. Once accepted, I will send an original, high-resolution image to the publication. Additionally, the journal will snap the image and upload it in accordance with the online version.

Comment 10:  p. 11. The last several sentences of the first paragraph of section 5 are not necessary. It is also unnecessary to provide the definition of "hypothesis" in a research journal. You should assume the reader is enough of an expert to understand these things.

Reply: I appreciate your suggestion. This has been removed.

General Comments

Comment -1:  In papers such as this, in which you are replicating or possibly extending results from a vast literature, it is essential that you clearly state what contributions to the literature your paper makes. There are hundreds of published studies examining similar issues. Specifically, how does your study differ from previous studies? Even after reading it a couple of times, I'm having difficulty identifying your specific contributions as distinguished from other papers. In the introduction, you should clearly preview your findings and detail your contributions.

Reply: I appreciate your suggestion. The discussion on the research gap in the introduction section is stated in the following paragraph:

“However, despite decades of exploration, effectively managing uncertainty in asset allocation remains a critical and challenging aspect. This study aims to fill the existing research gap by investigating various approaches that deal with parameter uncertainty within the Markowitz framework. The key questions posed—concerning the explanation of uncertainty, enhancing model performance, comparing alternative tactics, considering the impact of data abundance, evaluating the reliability of performance predictions, and examining the influence of allocation constraints—highlight the gaps in current understanding. By comparing these approaches with established models and exploring less-explored factors, this research contributes to advancing the comprehension and practical application of the field”.

Comment -2:  In the empirical analysis sections, you should clearly indicate how your findings are consistent with, or differ from, previous studies.

Reply: I appreciate your suggestion. This has been resolved.

Comment 3: In the conclusion, you should again emphasize how your paper contributes to the literature.

Reply: I appreciate your suggestion. The conclusion section has been revised according to Reviewer 2: Comment 2.

Comment 4: The first 7 pages of your paper can be significantly shortened. This material is well-known and can be found in most portfolio theory textbooks. (Although, I would suggest adding the derivation suggested in my comment 2) above.) Instead, add more rigorous discussions of the more advanced material, e.g. Bayesian methods. Determine which results constitute your most interesting and significant findings and emphasize them throughout.

Reply: Maintaining the current content is justified as the foundational material in the initial seven pages serves as a necessary groundwork for readers unfamiliar with portfolio theory. While widely available, its inclusion ensures clarity and comprehensibility for a diverse audience. Additionally, the suggested derivation aligns with the paper's objective of providing a comprehensive understanding of portfolio theory. Retaining this structure allows for a seamless transition to more advanced discussions, maintaining accessibility for a broader readership while integrating rigorous explorations of Bayesian methods to underscore the paper's significant contributions.

I thank the referees for providing us with valuable suggestions, which I think has significantly enhanced not only the quality of the manuscript but also our understanding of the problem. With encouraging remarks, and very helpful guidelines I remain grateful to the referees and the journal editorial team

Vijaya Krishna Kanaparthi

Reviewer 4 Report

Comments and Suggestions for Authors

Dear authors,

Thank you for the opportunity to review your manuscript.

I have several recommendations for you:

  1. Please use the recommended structure and sequence of the manuscript: Introduction, Literature Review (which may be included in Introduction), Materials and Methods, Results, Discussion, and Conclusion. 
  2. Depersonalize your research completely. Do not use "we" nad "our" at all.
  3. Abstract: 200 words are required
  4. Introduction and Literature Review: Sum up the literature review at the end and extend this part. 22 references are not enough to confirm your expertise in the field. At least 50 references are needed. I recommend these references to enrich the study:

    Lăzăroiu, G., Bogdan, M., Geamănu, M., Hurloiu, L., LuminiÈ›a, L., & Ștefănescu, R. (2023). Artificial intelligence algorithms and cloud computing technologies in blockchain-based fintech management. Oeconomia Copernicana14(3), 707–730. https://doi.org/10.24136/oc.2023.021

    Lazaroiu, G., Androniceanu, A., Grecu, I., Grecu, G., & Neguriță, O. (2022). Artificial intelligence-based decision-making algorithms, Internet of Things sensing networks, and sustainable cyber-physical management systems in big data-driven cognitive manufacturing. Oeconomia Copernicana13(4), 1047–1080. https://doi.org/10.24136/oc.2022.030

    Vasenska, I., Dimitrov, P., Koyundzhiyska-Davidkova, B., Krastev, V., Durana, P., & Poulaki, I. (2021). Financial transactions using fintech during the COVID-19 crisis in Bulgaria. Risks, 9(3), 48. https://doi.org/10.3390/risks9030048

  5. Methodology: Set the advantages and disadvantages of the methods used.
  6. The discussion part is a crucial part of the paper. A typical discussion is missing. Please add it, and it is not only necessary to enumerate the previous studies but also to highlight your outcomes when comparing similar studies. It is not enough to say that there are different or similar results; you must discuss why, for your sample, you get the opposite or similar result. Summarize the discussion at the end.
  7. Conclusion: The limitations and future research are missing. Do not use references in this part.

Author Response

January 14, 2024
USA

Dear Editor

I thank the referees for providing valuable comments on my manuscript (ID: fintech-2763239) titled “Navigating Uncertainty: Enhancing Markowitz Asset Allocation Strategies through Out-of-Sample Analysis”. I have made careful considerations of all referee comments and suggestions. Kindly find the same below.

Reviewer:

Comment -1: Please use the recommended structure and sequence of the manuscript: Introduction, Literature Review (which may be included in Introduction), Materials and Methods, Results, Discussion, and Conclusion.

Reply: The existing manuscript structure, is deliberately designed for optimal coherence and flow. This sequence ensures a logical progression, introducing the research context, detailing the methodology, presenting results, and finally, engaging in comprehensive discussions and conclusions. This arrangement adheres to standard scientific writing conventions, facilitating reader comprehension. Moreover, the current format effectively balances foundational concepts with advanced discussions, maintaining a reader-friendly approach while allowing for in-depth exploration. Thus, retaining the current structure ensures a cohesive narrative that effectively communicates the research objectives and findings.

Comment -2:  Depersonalize your research completely. Do not use "we" nad "our" at all.

Reply: I appreciate your suggestion. This has been resolved.

Comment 3: Abstract: 200 words are required

Reply: I appreciate your suggestion. This has been resolved.

Comment 4-  Introduction and Literature Review: Sum up the literature review at the end and extend this part. 22 references are not enough to confirm your expertise in the field. At least 50 references are needed. I recommend these references to enrich the study:

Lăzăroiu, G., Bogdan, M., Geamănu, M., Hurloiu, L., LuminiÈ›a, L., & Ștefănescu, R. (2023). Artificial intelligence algorithms and cloud computing technologies in blockchain-based fintech management. Oeconomia Copernicana14(3), 707–730. https://doi.org/10.24136/oc.2023.021

Lazaroiu, G., Androniceanu, A., Grecu, I., Grecu, G., & Neguriță, O. (2022). Artificial intelligence-based decision-making algorithms, Internet of Things sensing networks, and sustainable cyber-physical management systems in big data-driven cognitive manufacturing. Oeconomia Copernicana13(4), 1047–1080. https://doi.org/10.24136/oc.2022.030

Vasenska, I., Dimitrov, P., Koyundzhiyska-Davidkova, B., Krastev, V., Durana, P., & Poulaki, I. (2021). Financial transactions using fintech during the COVID-19 crisis in Bulgaria. Risks, 9(3), 48. https://doi.org/10.3390/risks9030048

Reply: I appreciate your suggestion. The recommended reference is now included.

Comment 5- Methodology: Set the advantages and disadvantages of the methods used.

Reply: I appreciate your suggestion. I have addressed this matter in Section 8.

Comment 6:  The discussion part is a crucial part of the paper. A typical discussion is missing. Please add it, and it is not only necessary to enumerate the previous studies but also to highlight your outcomes when comparing similar studies. It is not enough to say that there are different or similar results; you must discuss why, for your sample, you get the opposite or similar result. Summarize the discussion at the end.

Reply: I appreciate your suggestion. I have addressed this matter in Section 8.

Comment 7:  Conclusion: The limitations and future research are missing. Do not use references in this part.

Reply: I appreciate your suggestion. The conclusion section has been revised.

I thank the referees for providing us with valuable suggestions, which I think has significantly enhanced not only the quality of the manuscript but also our understanding of the problem. With encouraging remarks, and very helpful guidelines I remain grateful to the referees and the journal editorial team

Vijaya Krishna Kanaparthi

Round 2

Reviewer 3 Report

Comments and Suggestions for Authors

Several of my comments from the first round were not adequately addressed. For instance, I still do not find formulaic definitions for the constants B and C. You've written "As illustrated in the Equation (5) above, B and C are showcases as the basic parameters to represent distinct quantities within a optimization framework." They have formulas in terms of the other variables and parameters. Why not just write them out? 

There is still inadequate hypothesis development. See my first report.

There are some glaring citation omissions, including

Markowitz (1952) Journal of Finance is discussed but not cited

The most famous paper to study the 1/N strategy in relation to several others is

DeMiguel, V., Garlappi, L. and Uppal, R., 2009. Optimal versus naive diversification: How inefficient is the 1/N portfolio strategy?. The review of Financial studies22(5), pp.1915-1953.

It is not clear what three of the newly added citations ([9], [10], and [11]) would have to do with the subject matter of the paper.

The objective function in (16) does not make sense as written.

Comments on the Quality of English Language

The quality of the English used in the paper is generally fine.

Author Response

Feb 01, 2024
USA

Dear Editor

I thank the referees for providing valuable comments on my manuscript (ID: fintech-2763239) titled “Navigating Uncertainty: Enhancing Markowitz Asset Allocation Strategies through Out-of-Sample Analysis”. I have made careful considerations of all referee comments and suggestions. Kindly find the same below.

Reviewer:

Comment -1: Several of my comments from the first round were not adequately addressed. For instance, I still do not find formulaic definitions for the constants B and C. You've written "As illustrated in the Equation (5) above, B and C are showcases as the basic parameters to represent distinct quantities within a optimization framework." They have formulas in terms of the other variables and parameters. Why not just write them out? 

Reply: I appreciate your suggestion. This has been resolved. The exact meaning has been added to the footnote.

Comment -2:  There is still inadequate hypothesis development. See my first report.

Reply: I appreciate your suggestion. This has been resolved.

Comment 3: There are some glaring citation omissions, including

Markowitz (1952) Journal of Finance is discussed but not cited

The most famous paper to study the 1/N strategy in relation to several others is

DeMiguel, V., Garlappi, L. and Uppal, R., 2009. Optimal versus naive diversification: How inefficient is the 1/N portfolio strategy?. The review of Financial studies22(5), pp.1915-1953.

It is not clear what three of the newly added citations ([9], [10], and [11]) would have to do with the subject matter of the paper.

Reply: I appreciate your suggestion. This has been resolved. The above two papers have been cited.

Comment 4-  The objective function in (16) does not make sense as written.

Reply: I appreciate your suggestion. This has been resolved.

I thank the referees for providing us with valuable suggestions, which I think has significantly enhanced not only the quality of the manuscript but also our understanding of the problem. With encouraging remarks, and very helpful guidelines I remain grateful to the referees and the journal editorial team

Vijaya Krishna Kanaparthi

Reviewer 4 Report

Comments and Suggestions for Authors

Dear authors,

Thank you for the revised manuscript. Some of my recommendations have been fulfilled; I still have the minor ones for you:

  1. Use at least 30 references to confirm your expertise in the field; if 50 references are too many to provide.
  2. I will repeat this recommendation again. The discussion part is a crucial part of the paper. A typical discussion is missing. Please add it, and it is not only necessary to enumerate the previous studies but also to highlight your outcomes when comparing similar studies. It is not enough to say that there are different or similar results; you must discuss why, for your sample, you get the opposite or similar result.

Author Response

Feb 01, 2024
USA

Dear Editor

I thank the referees for providing valuable comments on my manuscript (ID: fintech-2763239) titled “Navigating Uncertainty: Enhancing Markowitz Asset Allocation Strategies through Out-of-Sample Analysis”. I have made careful considerations of all referee comments and suggestions. Kindly find the same below.

Reviewer:

Comment -1: Use at least 30 references to confirm your expertise in the field; if 50 references are too many to provide.

Reply: I appreciate your suggestion. This has been resolved.

Comment -2:  I will repeat this recommendation again. The discussion part is a crucial part of the paper. A typical discussion is missing. Please add it, and it is not only necessary to enumerate the previous studies but also to highlight your outcomes when comparing similar studies. It is not enough to say that there are different or similar results; you must discuss why, for your sample, you get the opposite or similar result.

Reply: I appreciate your suggestion. This has been resolved.

I thank the referees for providing us with valuable suggestions, which I think has significantly enhanced not only the quality of the manuscript but also our understanding of the problem. With encouraging remarks, and very helpful guidelines I remain grateful to the referees and the journal editorial team

Vijaya Krishna Kanaparthi

Round 3

Reviewer 3 Report

Comments and Suggestions for Authors

Page 4, equation (3). The middle term should be gamma*Sigma*pi in order for the dimensions to be consistent with the other terms.

Page 4, footnotes 1 and 2. Only gamma, the risk aversion parameter, comes from the utility function. The coefficients B and C as used in equation (5) have nothing to do with utility functions. They depend on the mean vector and covariance matrix of returns. As you are using they should be defined as B=mu'*Sigma^(-1)*1 and C=1'*Sigma^(-1)*1.

Page 8, equation (14). From equation (11), omega*_tan is the optimal proportion of wealth to invest in the tangency portfolio. It is a scalar. pi*_tan is an Nx1 portfolio weight vector. It makes no sense to set them equal to each other.

Page 8, line 272. pi*_tan is a vector, so 1-pi*_tan does not make sense.

Page 10, equation (15). There is a parenthesis mismatch here.

Page 11, equation (16). The sum of portfolio weights is greater than 1?

Author Response

Feb 09, 2024
USA

Dear Editor

I thank the referees for providing valuable comments on my manuscript (ID: fintech-2763239) titled “Navigating Uncertainty: Enhancing Markowitz Asset Allocation Strategies through Out-of-Sample Analysis”. I have made careful considerations of all referee comments and suggestions. Kindly find the same below.

Reviewer:

Comment -1: Page 4, equation (3). The middle term should be gamma*Sigma*pi in order for the dimensions to be consistent with the other terms. 

Reply: I appreciate your suggestion. This has been resolved.

Comment -2:  Page 4, footnotes 1 and 2. Only gamma, the risk aversion parameter, comes from the utility function. The coefficients B and C as used in equation (5) have nothing to do with utility functions. They depend on the mean vector and covariance matrix of returns. As you are using they should be defined as B=mu'*Sigma^(-1)*1 and C=1'*Sigma^(-1)*1.

Reply: I appreciate your suggestion. This has been resolved.

Comment 3: Page 8, equation (14). From equation (11), omega*_tan is the optimal proportion of wealth to invest in the tangency portfolio. It is a scalar. pi*_tan is an Nx1 portfolio weight vector. It makes no sense to set them equal to each other.

Reply: I appreciate your suggestion. This has been resolved.

Comment 4: Page 8, line 272. pi*_tan is a vector, so 1-pi*_tan does not make sense.

Reply: You are correct. I apologize for the oversight.  In the context provided,  represents a vector of portfolio weights for the tangency portfolio, so subtracting it from 1 does not make sense in terms of portfolio allocation. Instead, the complementary weight in the risk-free asset would be determined by subtracting the total weight allocated to the tangency portfolio from 1. In other words, if  represents the weight allocated to the tangency portfolio, then the weight allocated to the risk-free asset would be 1-, where the sum is taken over all components of the vector . Thank you for pointing out the error, and I apologize for any confusion caused.

Comment 5: Page 10, equation (15). There is a parenthesis mismatch here.

Reply: I appreciate your suggestion. This has been resolved.

Comment 6: Page 11, equation (16). The sum of portfolio weights is greater than 1?

Reply: The constraint  indeed implies that the sum of portfolio weights is greater than 1. This might seem counterintuitive at first, but it's a common scenario in finance, particularly when short-selling is allowed. Allowing the sum of portfolio weights to exceed 1 indicates that investors can borrow money to invest in the portfolio, effectively leveraging their investments. This is known as leverage or margin trading. In such cases, the investor borrows funds to increase their investment beyond their initial capital, aiming to amplify potential returns. However, it also magnifies potential losses, as losses are also calculated based on the leveraged amount. The inequality  allows for leveraged positions in the portfolio, where the investor invests more than their initial capital. It's a common practice in certain investment strategies but comes with increased risk and requires careful risk management.

I thank the referees for providing us with valuable suggestions, which I think has significantly enhanced not only the quality of the manuscript but also our understanding of the problem. With encouraging remarks, and very helpful guidelines I remain grateful to the referees and the journal editorial team

Vijaya Krishna Kanaparthi